# Dynamic Evolution and Collaborative Development Model of Urban Agglomeration in Hexi Corridor from the Perspective of Economic Flow

**Xiaoyi Xie [1] and Peiji Shi [1,2,*]**

1   College of Geography and Environmental Science, Northwest Normal University, Lanzhou 730070, China
2   Gansu Engineering Research Center of Land Utilization and Comprehension Consolidation, Northwest Normal University, Lanzhou 730070, China
*   Correspondence: shipj@nwnu.edu.cn; Tel.: +86-138-9366-5158

**Abstract:** Under the green goals of the carbon peak and carbon neutrality, understanding how to develop the economy with high quality is an important issue facing regional development. Based on the years 2000, 2010, and 2020, this paper studies the industrial function connection path and economic network characteristics of the Hexi Corridor through an urban flow model, dominant flow analysis, modified gravity model, and social network analysis method, and puts forward an economic synergistic development model. It is of great significance to strengthen the urban connection in the Hexi Corridor and give full play to the overall competitive advantage. The results are as follows. (1) The overall function of the urban agglomeration is weak, the outward function of manufacturing is outstanding, the complementary network is highly complicated and evolving, and the environment and public service and tourism industry have apparent advantages. (2) The backbone correlation axes of the "three industries" show the characteristics of a closed triangular connection, dual-core linkage development, and multi-center multi-axis interaction. (3) The economic network has a greater agglomeration effect than diffusion effect, with prominent grouping characteristics, forming a network structure of "one man, three vices, and many nodes" and a significant spatial proximity effect. (4) Based on geographical proximity, the "one axis, four circles, multiple points, and multiple channels" synergistic development model, which breaks administrative barriers, becomes the endogenous driving force for the evolution of the economic network.

**Keywords:** Hexi Corridor urban agglomeration; industrial development axis; economic network; regional coordinated development



## 1. Introduction

Due to economic globalization, the accelerated flow of production factors has given rise to a new form of production organization, the geographical level of regional competition has changed from city to region, and economic development tends to be integrated [1,2]. As essential support for the unimpeded and revitalized Silk Road Economic Belt, the Hexi Corridor city cluster has a prominent ecological position, good resource endowment, and apparent advantages in geographical channels. It is a crucial link in promoting the development of the western region [3]. However, the distribution of oases in the belt geography space determines the cluster-like economic, spatial pattern of the urban agglomeration, with the oasis center towns being mainly and relatively independent. At the same time, constrained by resources and the ecological environment, topography, geomorphology, and economic infrastructure conditions, there are problems such as the weak competitive ability of industrial clusters, the low degree of internal and external opening, and slow integrated and coordinated development [4]. Under the green goal of achieving the carbon peak and carbon neutralization, and the vision of "One Belt and One Road," it is of great significance to guide the close and orderly economic linkage of urban agglomeration and

explore the high-quality coordinated development mode of complementary advantages, which enhance the urban connection of the Hexi Corridor, cultivate advanced industrial clusters, and give play to the overall competitive advantages [5].

Economic linkage is a correlative and participatory economic behavior based on the exchange of capital, technology, commodities, and other factors between cities, expressed as the interaction between regional economic entities [6]. Under the trend of globalization and informatization, the "space–time compression" effect has revolutionized the function and spatial pattern of cities, and the content of economic linkage research has been extended with the innovation of urban research theories and methods. Christaller's central place theory, Francois Perroux's growth pole theory, Ullman's spatial interaction theory, and Friedmann's core-edge theory laid the foundation for regional economic linkages. From a content point of view, in the 1930s, western scholars began to focus on urban spatial economic linkages [7]. Zipf first applied the law of gravity to urban spatial interactions, and Converse used the rupture point model to delineate urban spatial economic linkages [8]. Since then, many scholars have conducted quantitative analysis and empirical studies on the economic linkages between cities and their hinterlands based on the gravity model. In the 1950s and 1960s, western societies experienced post-war reconstruction, and the phenomenon of urban agglomeration territorialization emerged. Isard applied input–output analysis to study the spatial linkage of regional hinterland economies, and Wu used the CA model to investigate the polycentricity structure of agglomeration linkages in urban agglomerations. Friedmann further pointed out that a high concentration of regional productivity is the greatest driving force for the formation of urban agglomerations, opening up the economic analysis of urban agglomerations [9]. In the 1970s and 1980s, due to the increase in land prices in urban centers and the development of modern transportation, urban manufacturing and service industries moved to the suburbs, and urban economic networks were mainly constructed through industrial linkages and enterprise networks. Henderson and Meyer built the equilibrium model of the city scale and the dynamic model of the city system, which promoted the theoretical model of urban economic linkage from static analysis to dynamic [7,10]. The development of modern transportation and communication networks has led to the formation of open and complex networks in urban systems, and Castells' spatial flow research has become a hotspot. Since the 1990s, foreign scholars have generally studied urban spatial economic linkages from the perspective of transportation networks and enterprise spatial organization networks. Goet studied the spatial characteristics of urban economic linkages from the perspective of transportation networks. Burger studied the impact of large corporate organizational networks on urban, regional, and even global linkages from a micro-organizational perspective [11]. From a content point of view, scholars mainly study economic connection networks from two directions: (1) the first direction involves describing the direction of economic connection under the background of big data, such as human flow, logistics, and information flow [12–14]; (2) regarding the second direction, based on the statistical data of urban economic flow, such as transportation networks, infrastructure networks, industrial correlations, and enterprise spatial organization, the economic connection network is constructed using the interlocking model, gravity model, and urban flow model. The network characteristics are explored by using social network analysis [15–24]. Although the significant data stream can accurately identify the linkage direction of factors, it is difficult to comprehensively reflect the change in urban economic linkages. Economic flow methods such as the gravity model start from the macroeconomic level, but it is difficult to reflect the fundamental spatial vector characteristics of economic relations between cities [25]. In addition, the selection of research regions focuses on global cities, large urban agglomerations, or other areas with high economic levels, with less research conducted on less developed regions with development potential. Moreover, the research focuses on the intensity of economic linkages and spatial patterns, while the portrayal of urban functional collaboration is relatively vague [26].

To summarize, this paper chooses the Hexi Corridor urban agglomeration as the research area, which is at the forefront of the opening up of the "Belt and Road" and located

in an economically underdeveloped region with an extremely important geographical location. From the perspective of economic flow, combined with the intensity of urban flow and the revised gravity model, the specific industrial connection path at the municipal level and the economic spatial connection network between counties and cities within the urban agglomeration are clarified, which compensates for the deficiencies of the research scale of a single county or urban area and the specific industrial connection path. Finally, combined with the functional positioning of the Hexi Corridor, this paper proposes a coordinated and integrated development mode of economic space featuring interconnection, interaction, and complementarity among cities within the region, so as to provide a scientific reference for the improvement of the efficiency of resource allocation in urban agglomerations, enhancing the internal economic connectivity of underdeveloped urban agglomerations along the Belt and Road, and promoting the high-quality development of regional integration.

## 2. Study Area and Methods

### 2.1. Study Area

The Hexi Corridor is connected with Qinghai in the south by the Qilian Mountains and Inner Mongolia in the north by the Beishan Mountain system. The terrain is high in the south and west, and low in the north and east, forming a topography of two mountains sandwiched by a plain [27]. The Gobi Desert is widely distributed. There are three Gobi oases with internal flowing water systems from east to west: the Shiyang River, Heihe River, and Shule River. The population mainly concentrates and distributes along the oasis area, and irrigation agriculture is well developed [28,29]. The total amount of water resources is 6.076 billion m, surface water resources are 5.447 billion, and groundwater resources are 5.923 billion m. The climate type is mainly a temperate continental arid climate, and the ecological status is prominent. The Hexi Corridor urban agglomeration has a long history, a unique region, and profound cultural deposits. It is a critical urban area in Gansu Province and belongs to the growing urban agglomeration. According to "The Opinions of Gansu Provincial Committee of the Communist Party of China and Gansu Provincial People's Government on Improving Regional Development Layout and Cultivating New Economic Growth Points and Growth Poles and Growth Belts (Ganfa [2020] No. 7)" and the "14th Five-Year Plan", the Economic Belt Development Plan of the Hexi Corridor, it is determined that the urban agglomeration of the Hexi Corridor includes 20 counties and districts under the jurisdiction of five cities, namely Jiayuguan, Wuwei, Zhangye, Jinchang, and Jiuquan(Figure 1). The total area is 247,800 square kilometers, accounting for 57.19% of the province.

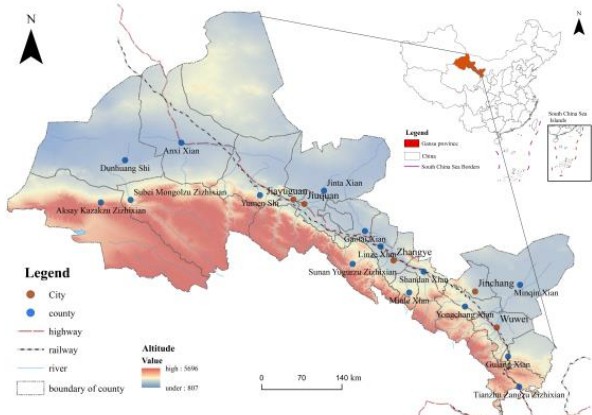

**Figure 1.** Research scope.

### 2.2. Methods

This study uses an urban flow model and dominant flow analysis to study the industrial connection path and urban function connection of the urban agglomeration at the municipal level, as well as the gravity model and social network analysis to analyze the

spatial characteristics of the economic network at the county level, and analyzes the dynamic evolution of the economic connection network from multiple perspectives (Figure 2).

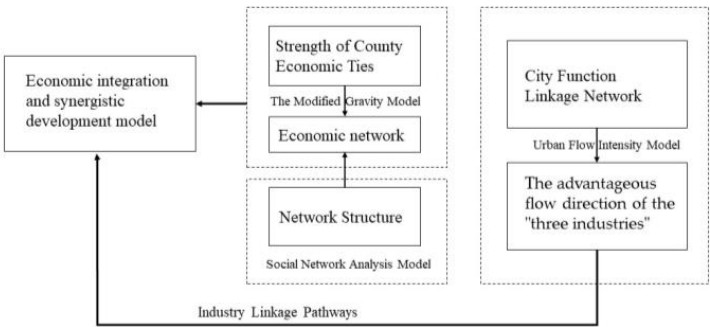

**Figure 2.** Research logic diagram.

2.2.1. Urban Flow Intensity Model

The urban flow model calculates the factor flow intensity in the economic agglomeration and diffusion activities between cities to reflect the ability of cities to communicate with the outside world. Based on the regional characteristics of the developed oasis agriculture in the Hexi Corridor and the current the research results, 18 industrial sectors were selected for research according to the "Industry Classification of National Economy (GB/T 4754-2017)" (Table 1) [28–31]. The formula is as follows:

$$F_i = E_i \times N_i = E_i \times (GDP_i/G_i) = GDP_i \times E_i/G_i = GDP_i \times k_i \tag{1}$$

where $F_i$ represents the intensity of urban flow in city $i$; $E_i$ represents the sum of the outward functions of city $i$; $N_i$ represents the efficiency of the outward functional volume of city $i$; $G_i$ represents the total number of employees in the city $i$; $GDP_i$ represents the per capita GDP of employees in units of the city $i$; $K_i$ represents the degree of urban flow tendency. The greater the $K_i$, the closer the relationship between city $i$ and other cities. Whether a city has an extroverted function $E_i$ depends on the location quotient of all departments. The formula for calculating the location quotient $Lq_{ik}$ of the $k$ industry sector in city $i$ is as follows:

$$Lq_{ik} = \frac{(G_{ik}/G_i)}{(G_k/G)}, (i = 1, 2, 3 \dots, n; k = 1, 2, \dots m) \tag{2}$$

where $n$ represents the number of cities; $m$ represents the number of industrial sectors; $G_{ik}$ represents the number of employees in the $k$ industry sector of city $i$; $G_{ik}$ represents the number of employees of the $k$ industry sector in China; $G_k$ represents the total number of employees in China. If $Lq_{ik} \leq 1$, $E_{ik} = 0$. If $Lq_{ik} > 1$, department $k$ of the city $i$ has external functions, $E_{ik} = G_{ik} - G_{ik}/Lq_{ik}$.

**Table 1.** Industrial sectors of the urban economy.

| Industry | Department | Function |
|---|---|---|
| Primary industry | 1. Agriculture, forestry, animal husbandry, and fishery | |
| Secondary industry | 2. Extractive industry; 3. manufacturing industry; 4. electricity, gas, and water production and supply industries; 5. construction | Urban productive function |

**Table 1.** *Cont.*

| Industry | Department | Function |
|---|---|---|
| The tertiary industry | 6. Transportation, warehousing, and postal services; 7. wholesale and retail trade; 8. finance industry; 9. real estate industry; 10. education; 11. health, social insurance, and social welfare; 12. culture, sports, and entertainment; 13. scientific research, technical services, and geological survey; 14. accommodation and catering industry; 15. information transmission, computer, service, and software industries; 16. leasing and commercial services; 17. water, environmental, and public utility management; 18. residential services, repairs, and other services | Urban service function |

### 2.2.2. Urban Functional Complementarity Index

This paper uses Formula (2) $Lq_{ik}$ to calculate the urban functional complementarity index ($FCI_{ij}$) and define the inter-urban industrial functional complementarity index ($FCI_k$) to reflect the $k$ industrial functions of the urban agglomeration [9]. The calculation formula is as follows:

$$FCI_{ij} = \sum_{k=1}^{n} \left| Lq_{ik} - Lq_{jk} \right| \tag{3}$$

$$FCI_k = \left| Lq_{ik} - Lq_{jk} \right| \tag{4}$$

### 2.2.3. Advantage Flow Analysis Method

Dominant flow analysis is vital for studying the correlation axis and regional spatial structure [25]. This paper identifies the first, second, third, and fourth dominant flow, determines the core backbone correlation axis of the three industries in the Hexi Corridor, and clarifies the urban function connection at the municipal level.

### 2.2.4. The Modified Economic Gravity Model

The gravity model is a standard method to measure the intercity economic connection [32]. We modified the gravity model from city quality, distance, and connection direction [33]. By referring to the "China Regional Development Evaluation Index System" issued by the "National Forum for Coordinated Regional Development Strategy (2021)" and the "Research Report of China City Quality Development Index (2022)" by the Center for Public Policy Research of Peking University, a total of 17 indicators from the two aspects of development strength and development potential were selected to evaluate the quality of the urban economy. SPSS principal component analysis was used to calculate the weight (Table 2) [30]. Economic distance considering time and economic cost was adopted instead of the traditional geographical straight-line distance. A coefficient $K_{ij}$ was added to the original model to distinguish the output and reception of economic links between cities to build the directed economic network of the urban agglomeration. After modification, the formula of the gravity model was as follows:

$$R_{ij} = k_{ij} \frac{M_i \times M_j}{d_{ij}{}^2}, R_{ij} = R_{ij_i} + R_{ij_j}, k_{ij} = \frac{M_i}{M_i + M_j} \tag{5}$$

$$d_{ij} = \sum_{\alpha=1}^{n} (\beta_\alpha C_\alpha T_\alpha)_{ij}, (\alpha = 1, 2) \tag{6}$$

$$\beta_\alpha = \frac{1}{2} \left( \frac{Passenger\ volume\ of\ mode\ a}{Total\ passenger\ capacity} + \frac{Freight\ volume\ of\ mode\ a}{Gross\ cargo\ volume} \right) \tag{7}$$

$R_{ij}$ represents the economic effect intensity of city $i$ on city $j$; $M_i$ and $M_j$ represent the economic quality of city $i$ and city $j$; $d_{ij}$ represents the economic distance between the two

places; $\beta_\alpha$ represents the $\alpha$ type of weights of transportation; $C_\alpha$ represents the $\alpha$ type of economic cost of transportation, and the economic cost of transportation by road and rail calculated according to 2:1 [30]; $T_\alpha$ represents the $\alpha$ type of the time cost of transportation.

**Table 2.** Index system and weight of the urban comprehensive economic level.

| First-Order Subsystem | Second-Order Subsystem | Indicator Layer | Weight |
|---|---|---|---|
| Strength of economic development | Aggregate of economy | Gross regional product | 0.076 |
| | | Rate of urbanization | 0.063 |
| | | Total fixed asset investment | 0.084 |
| | | Total retail sales of consumer goods | 0.069 |
| | Structure of economy | The proportion of the value added by the secondary industry in GDP | 0.018 |
| | | The proportion of the value added by the tertiary industry in GDP | 0.026 |
| | | Share of local fiscal revenue in GDP | 0.078 |
| | Economic benefits | Per capita savings balance at the end of the year | 0.077 |
| | | Per capita disposable income of urban residents | 0.055 |
| Economic development potential | Basic services | Local fiscal expenditure | 0.025 |
| | | Transportation, warehousing, and postal industry added value | 0.068 |
| | | Number of Internet broadband access users | 0.064 |
| | Science, education, culture and health services | Total number of hospital beds | 0.028 |
| | | Number of students enrolled in ordinary middle schools | 0.067 |
| | | Public library holdings | 0.063 |
| | Size of population | Total population at year-end | 0.058 |
| | | Urban population | 0.080 |

### 2.2.5. Social Network Analysis Model

Social network analysis is widely used in studying complex network relationships in economics and geography. In this paper, the social network analysis software UCINET 6.2 was used to analyze the network density, centrality, and cohesive subgroups of the economic connection network of the urban agglomeration. The specific calculation formula is as follows [34].

- Network Density

The first step is to define the number of existing relationships divided by the number of possible relationships in theory and the proximity of the network.

$$D = 2m/[n(n-1)] \tag{8}$$

where $D$ represents the network density value; $m$ represents the actual relation number; $n$ represents the maximum number of relationships in theory.

- Centrality of Network

(1)  Degree centrality (DC) is divided into out-degree and in-degree. Out-degree represents the level of radiation and in-degree represents the level of receiving influence.

$$C_{(out)i} = \frac{d_{(out)i}}{k-1}, C_{(in)i} = \frac{d_{(in)i}}{k-1} \tag{9}$$

where $C_{(out)i}$ represents the exit degree of node $i$; $C_{(in)i}$ represents the entry degree of node $i$; $d_{(out)i}$ represents the number of contacts issued by node $i$;

$d_{(in)i}$ represents the number of connections accepted. $k - 1$ represents the number of links between a point and the outside world when the number of subjects is $k$.

(2) Closeness centrality (CC) measures the proximity of a city's contact distance. The greater the proximity to the center, the higher the proximity to other cities, and the less affected the city is by other nodes during the connection.

$$C_{c(i)} = \left[ \sum_{i=1}^{n} d_{ij(c)} \right]^{-1} \tag{10}$$

where $C_{c(i)}$ represents the proximity to the center of node $i$; $d_{ij(c)}$ represents the economic distance between nodes $i$ and $j$.

(3) Betweenness centrality (BC) measures the "medium" function of node connections.

$$C_{b(i)} = \frac{1}{(N-1)(N-2)} \sum_{j \neq k} \frac{g_{jk(i)}}{g_{jk}} \tag{11}$$

where $C_{(b)i}$ represents the intermediate centrality of node $i$; $g_{jk}$ represents the number of shortest paths between points $j$ and $k$; $g_{jk(i)}$ represents the number of shortest paths passing through node $i$ between node $j$ and node $k$; $N$ indicates the number of nodes in the entire network.

- Cohesive Subgroups

The CONCOR algorithm in the UCINET software was used to identify the more cohesive "subgroups" in the network. The possibility of economic exchange and industrial transfer is more significant in the subgroup, which is conducive to guiding the regional economy's benign development from the whole network's perspective.

## 3. Results

### 3.1. City Industrial Functional Complementarity Analysis

The industrial functional complementarity index of the Hexi Corridor urban agglomeration is shown in Figure 3. The overall industrial functional complementarity network presented a complicated and compact trend during the study period. The industrial development was characterized by a high degree of agricultural specialization, steady improvement of industrialization, rapid development of the tertiary industry, and dislocated development of profitable industries in various regions.

From 2000 to 2010, the industrial base was improved, and vital complementary industries increased and showed similar characteristics. Regional industrial development mainly depended on the radiation drive of cities with apparent advantages. Jiuquan, Zhangye, Wuwei, and Jiayuguan have strong complementary connections among various industrial sectors and a solid industrial foundation. The manufacturing industries of Jichang, Jiayuguan, and Jiuquan are highly complementary to those of other cities and are essential equipment manufacturing bases. Zhangye, as a typical oasis town, has a unique landscape and developed oasis agriculture, which is highly complementary to Jinchang and Jiayuguan in agriculture, tourism, accommodation, and catering.

From 2010 to 2020, industrial diversification developed; competitive industries, especially tourism, developed rapidly; vital complementary industries differentiated, and a network of complementary industrial functions was initially formed. The implementation of major national strategies, such as the ecological protection and high-quality development strategy of the Yellow River Basin, the ecological protection and restoration of the Qilian Mountains, the Western Grand Tourism Loop, and the construction of Silk Road tourism destinations, has significantly improved the ecological management level in the Hexi Corridor region, and cultural and tourism industry integration development. Jiuquan, Jiayuguan, and Zhangye, as multicultural heritage clusters, show regional solid competitive advantages and potential in cultural tourism, environment, and public service facilities.

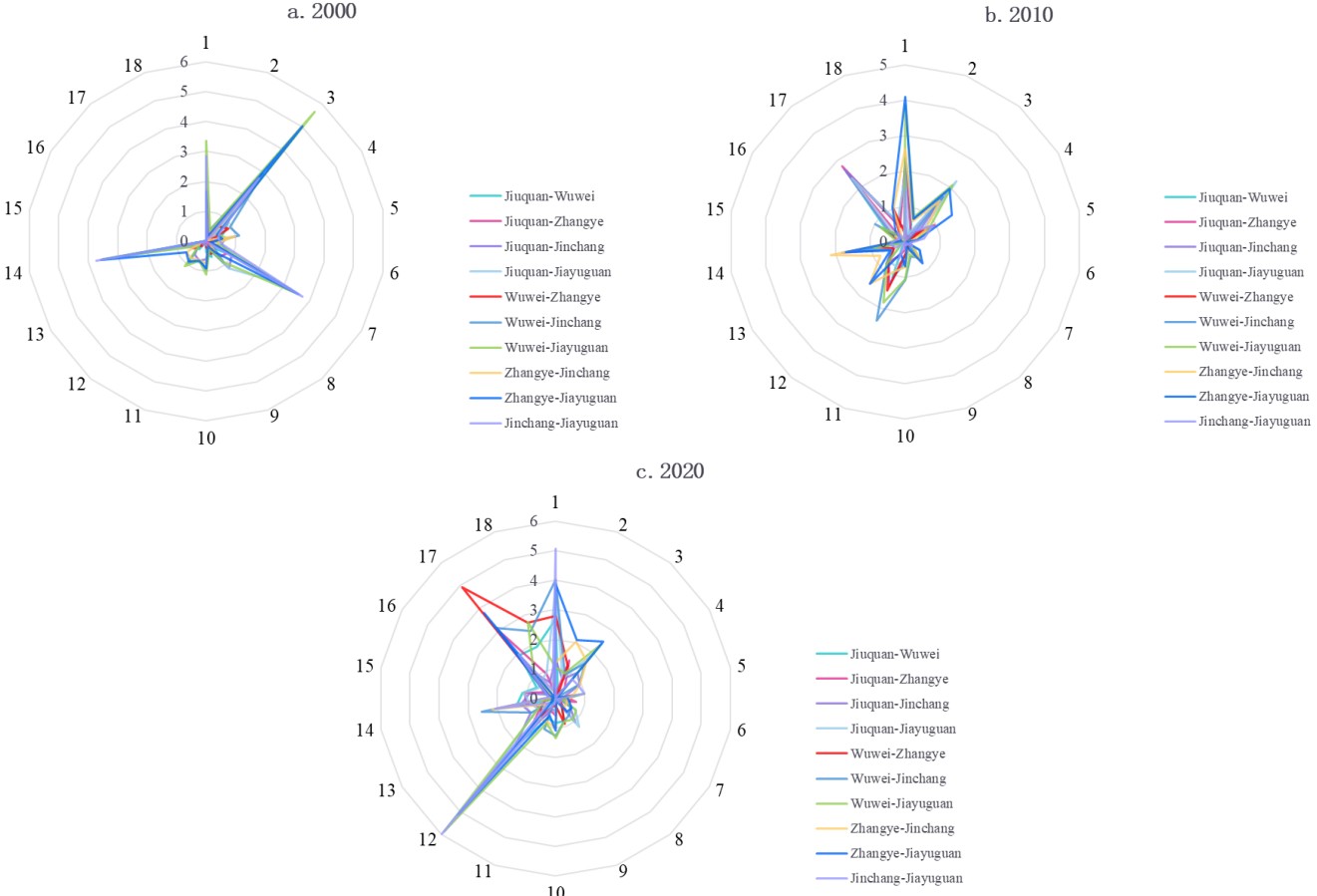

1 to 18 correspond to the industry sector in Table 1.

**Figure 3.** Industrial functional complementarity index in 2000, 2010, and 2020.

### 3.2. Urban Outward Function and Correlation Axis Analysis

The urban flow intensity model was used to analyze the outward function of industrial sectors in the urban agglomeration in 2020 (Table 3). The results show that the overall degree of outgoing service function in the urban agglomeration is low, but that the outgoing function of the manufacturing industry is far more than that of other industries. The tertiary industry has no obvious advantage, which reflects the high proportion of heavy industry in the urban agglomeration, and the transportation infrastructure needs to be improved. The weak flow ability of capital, technology, talent, and information leads to the weak interaction between commerce and high-tech industries, which, to some extent, hinders the high-quality development of the urban agglomeration and weakens the regional competitive advantages.

According to the structural decomposition of the urban flow intensity of specific cities (Table 4) and further exploration of the urban outward function, it can be seen that the quantities of outward function of the five cities in the Hexi Corridor are all greater than one, and the gap is small. The gap in outward radiation capacity is not significant. Jiuquan City has the most extensive urban flow intensity and a high urban flow tendency, which expands the flow range of factors and drives regional development. Wuwei and Zhangye have the second-highest urban flow intensity and the lowest urban flow tendency. Therefore, it is necessary to adjust and upgrade the industrial structure, improve the technical level, and enhance the city's comprehensive strength and radiation function. Jinchang City has a low-intensity urban flow and a high tendency towards urban flow. Although it has an excellent industrial base and advantages in the manufacturing industry, it has a weak influence on external radiation, which requires further improvement in resource integration and regional radiation-driving ability. Due to its small economic size, Jiayuguan City has

a lower total amount of factor flow than other cities. However, the urban flow tendency is the largest, and the influence of external service sectors such as manufacturing, culture, and tourism is strong. The potential of external connection between cities is enormous.

**Table 3.** Urban agglomeration total export capacity by industry.

| Industry | $E_k$ | Industry | $E_k$ |
|---|---|---|---|
| Agriculture, forestry, animal husbandry, and fishery | 0.71 | Education industry | 1.37 |
| Extractive industry | 0.21 | Health, social insurance, and social welfare | 0.49 |
| Manufacturing | 3.67 | Culture, sports, and entertainment | 1.06 |
| Electricity, gas, and water production and supply industries | 0.12 | Scientific research, technical services, and geological surveys | 0.18 |
| Construction industry | 0.89 | Accommodation and catering | 0.38 |
| Transportation, warehousing, and postal services | 0.67 | Information transmission, computers, services, and software | 0.13 |
| Wholesale and retail trade | 0.15 | Leasing and business services | 0.09 |
| Finance industry | 1.09 | Water conservancy, environment, and public facilities management | 1.34 |
| Real estate industry | 0.02 | Resident services, repairs, and other services | 0.12 |

**Table 4.** Cities' total outward energy and the degree and intensity of urban flow.

| | Urban Outward Function Quantity $E$ | Urban Functional Efficiency $N$ | The Intensity of Urban Flow $F$ | The Propensity of Urban Flow $K$ | GDP (100 Million Yuan) |
|---|---|---|---|---|---|
| Zhangye | 3.06 | 46.50 | 142.29 | 0.23 | 448.73 |
| Wuwei | 4.03 | 38.32 | 154.43 | 0.32 | 488.46 |
| Jiuquan | 4.20 | 42.32 | 177.74 | 0.40 | 618.22 |
| Jinchang | 3.09 | 37.19 | 114.92 | 0.34 | 340.31 |
| Jiayuguan | 2.45 | 48.33 | 118.41 | 0.42 | 283.40 |

The dominant flow method is used to analyze the intensity of the urban flow of the "three industries" in each city, identify the industrial core backbone connection axis, and obtain the three industries' first, second, third, and fourth most dominant flow directions (Figure 4). The dominant flow of the primary industry is dominated by the connection between Jiuquan and Zhangye, forming a closed triangular connection between Jiuquan, Zhangye, and Jinchang. The Heihe River and Shule River systems form a large oasis in Jiuquan and Zhangye, with good soil and water conditions. It has a crucial modern cold and drought agricultural area in the Hexi Corridor area. The good flow of the secondary industry forms a dual-core linkage pattern of wine and Jiashan and Jinwu. With Jiayuguan and Jiuquan as the lead and Jiuquan–Zhangye and Jinchuan–Wuwei as the backbone axes, the tertiary industry has evolved into a multi-center and multi-axis interactive pattern. Zhangye City undertakes the function of a "transit station", which, on the one hand, receives the functional radiation of Jiuquan–Jiayuguan, the dominant city in the western section of the Hexi Corridor.

On the other hand, it is connected with the neighboring Jinchang City in terms of material exchange at the level of production and consumption, connecting the belt industrial connection path as a network. Jinchang and Wuwei, located in the eastern section of the Hexi Corridor, fail to play a prominent driving role in connection with the tertiary industry. Jinchang has a weak foundation for the development of the tertiary industry. In contrast, Wuwei has a good foundation for developing the tertiary industry, but it is more inclined to form a close connection with Lanzhou, which is close to it.

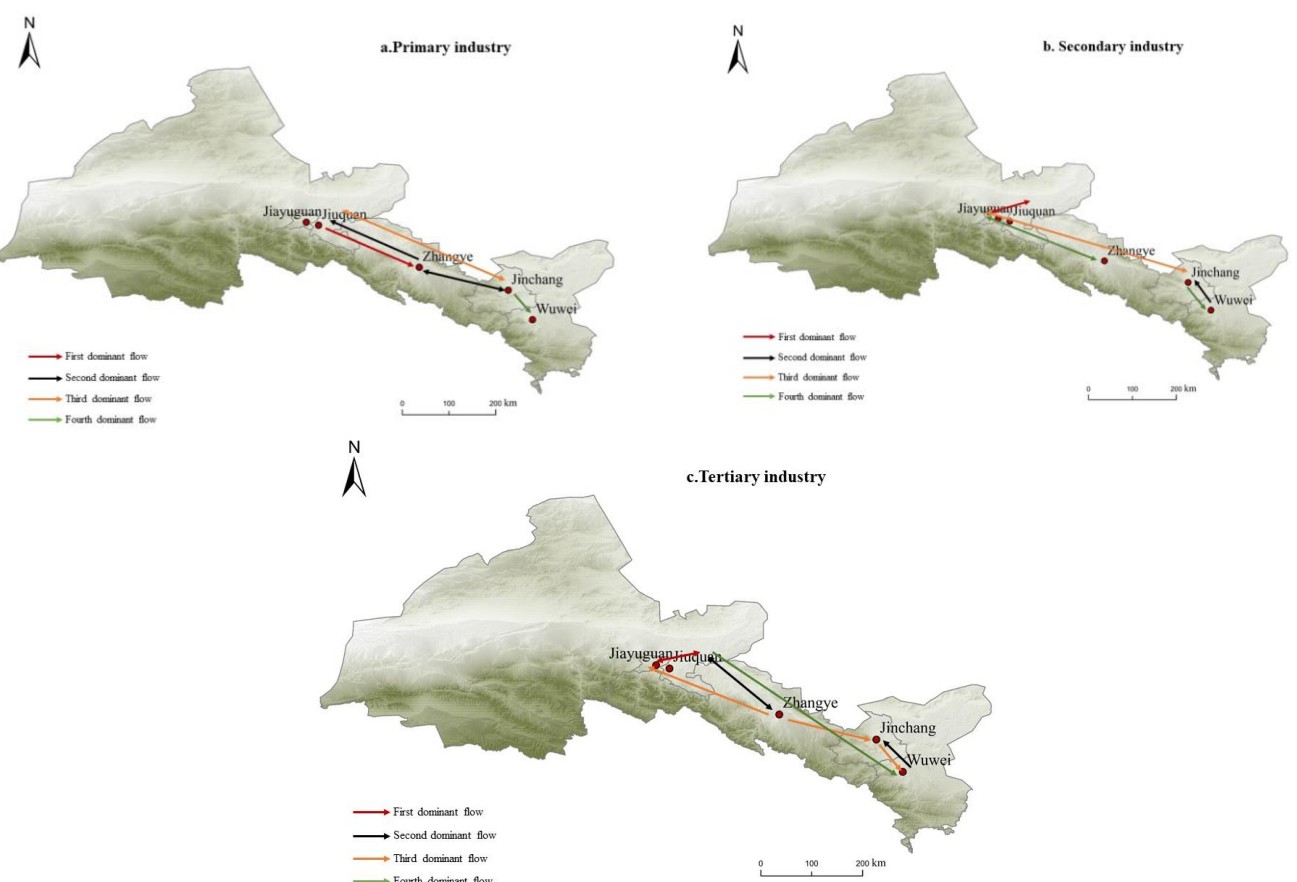

**Figure 4.** The advantageous flow direction of the "three industries" in the urban agglomeration in 2020.

### 3.3. Space–Time Evolutionary Analysis of Economic Network

The intensity of inter-county economic ties was calculated according to the modified gravity model, and the structure chart of the economic network in 2000, 2010, and 2020 was drawn (Figure 5). From the perspective of spatial evolution, the economic links of urban agglomerations show the distribution characteristics of clusters, forming the convergence development pattern of four economic clusters: Jiuquan–Jiayuguan, Zhangye, Jinchang–Wuwei and Dunhuang–Guazhou. The grouped economic links show a spatial solid proximity effect, where the links within groups are close, but the links between groups are weak. As the central town of the Zhangye grouping, Ganzhou District only shows a robust third-level connection with Linze County, which is close to it, and the other groups are connected with ordinary fourth-level strength. As Aksai, Subei, and Sunan are located at the edge of the urban agglomeration and have a long spatial distance from other counties, they have not yet effectively joined the economic connection network of the urban agglomeration.

In terms of the time evolution, the economic linkage strength of urban agglomerations has increased significantly, with the top 20 economic linkages increasing from 188.46 in 2000 to 1305.89 in 2020, increasing more than six times (Table 5). The integration of Jiujia, Jinwu, and intercity high-grade highway integration construction has effectively promoted cross-administrative factor flow and increasingly frequent economic interaction between cities. However, the polarization between counties and districts is increasing, with the extreme difference increasing from 65.38 in 2000 to 515.02 in 2020. The intensity of economic ties in areas not open to railroads at the edges of urban clusters such as Aksai, Subei, and Sunan Counties has been in the back row, with the leapfrog development of backward areas being restricted.

**Table 5.** The top 20 pairs of two-party relationships in the economic connection strength of the Hexi Corridor urban agglomeration.

| In 2000 | | In 2010 | | In 2020 | |
|---|---|---|---|---|---|
| Pt$_e$ | Value | Pt$_e$ | Value | Pt$_e$ | Value |
| Suzhou District–Jiayuguan | 67.28 | Jiayuguan–Suzhou District | 158.67 | Jiayuguan–Suzhou District | 528.62 |
| Liangzhou District–Gulang County | 20.43 | Liangzhou District–Gulang County | 64.27 | Liangzhou District–Suzhou District | 129.47 |
| Suzhou District–Kinta County | 18.26 | Liangzhou District–Yongchang County | 32.36 | Liangzhou District–Yongchang County | 91.88 |
| Liangzhou District–Jinchuan District | 15.53 | Liangzhou District–Jinchuan District | 32.09 | Liangzhou District–Jinchuan District | 88.58 |
| Liangzhou District–Yongchang County | 13.12 | Suzhou District–Kinta County | 28.11 | Suzhou District–Kinta County | 75.87 |
| Ganzhou District–Linze County | 8.65 | Ganzhou District–Linze County | 23.58 | Ganzhou District–Linze County | 53.20 |
| Jinchuan District–Yongchang County | 8.26 | Jinchuan District–Yongchang County | 13.39 | Jinchuan District–Yongchang County | 45.73 |
| Jiayuguan–Jinta County | 5.83 | Jiayuguan–Yumen City | 12.43 | Jiayuguan–Yumen City | 41.13 |
| Liangzhou District–Tianzhu County | 4.15 | Jiayuguan–Jinta County | 11.73 | Jiayuguan–Jinta County | 34.98 |
| Liangzhou District–Minqin County | 3.64 | Liangzhou District–Tianzhu County | 10.41 | Liangzhou District–Minqin County | 30.83 |
| Jiayuguan–Yumen City | 3.16 | Liangzhou District–Minqin County | 10.00 | Liangzhou District–Tianzhu County | 26.93 |
| Ganzhou District–Gaotai County | 2.77 | Ganzhou District–Minle County | 8.10 | Suzhou District–Yumen City | 22.51 |
| Suzhou District–Yumen City | 2.56 | Suzhou District–Yumen City | 7.80 | Jiayuguan–Gaotai County | 19.57 |
| Ganzhou District–Minle County | 2.40 | Ganzhou District–Gaotai County | 7.33 | Ganzhou District–Gaotai County | 18.39 |
| Liangzhou District–Shandan County | 2.27 | Liangzhou District–Shandan County | 6.34 | Liangzhou District–Shandan County | 17.75 |
| Suzhou District–Ganzhou District | 2.17 | Jiayuguan–Ganzhou District | 5.81 | Ganzhou District–Minle County | 17.30 |
| Suzhou District–Gaotai County | 2.13 | Jiayuguan–Gaotai County | 5.46 | Jiayuguan–Ganzhou District | 17.14 |
| Jiayuguan–Ganzhou District | 2.00 | Jiayuguan–Linze County | 5.13 | Jiayuguan–Linze County | 16.59 |
| Suzhou District–Linze County | 1.93 | Ganzhou District–Shandan County | 4.96 | Suzhou District–Gaotai County | 15.82 |
| Jiayuguan–Gaotai County | 1.90 | Suzhou District–Gaotai County | 4.85 | Suzhou District–Linze County | 13.60 |
| The sum | 1305.89 | The sum | 452.79 | The sum | 188.46 |

Pt$_e$ refers to pairs of two-party relationships in the economic connection.

### 3.4. Economic Network Topology Analysis

By applying SNA, we constructed an adjacency matrix with the economic connection strength of node towns to analyze the structure and overall characteristics of the economic spatial correlation network (Figure 6) [35,36]. During the study period, the economic network structure of the urban agglomeration tends to be complicated and dense. From the network framework, the economic links are distributed and decentralized, with Suzhou District, Jiayuguan City, Liangzhou District, and Ganzhou District as the central nodes. Unlike the network structure of cluster-like cities, the economic network of the Hexi Corridor belongs to the belt-like form and the geographical space. The central node cities are the critical paths for the other node cities to conduct economic exchanges and have a solid ability to regulate the resource factors. Concerning the economic linkage pattern (Figure 5), the city cluster forms a "one main, three vice, multi-node" economic and spatial network structure, with Jiuquan and Jiayuguan as the main centers, Jinwu, Zhangye and Dungua as sub-centers, and strong counties such as Yumen, Gaotai, Minle, Gulang, and Minqin as multi-nodes.

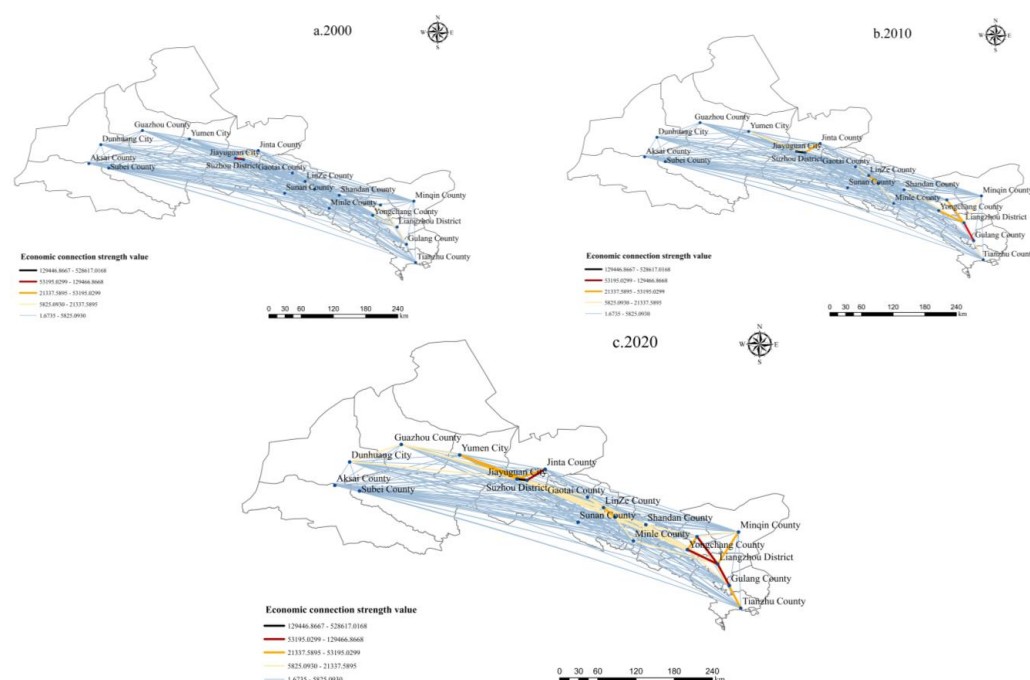

**Figure 5.** The evolution of the economic connection pattern of the Hexi Corridor urban agglomeration in 2000, 2010, and 2020.

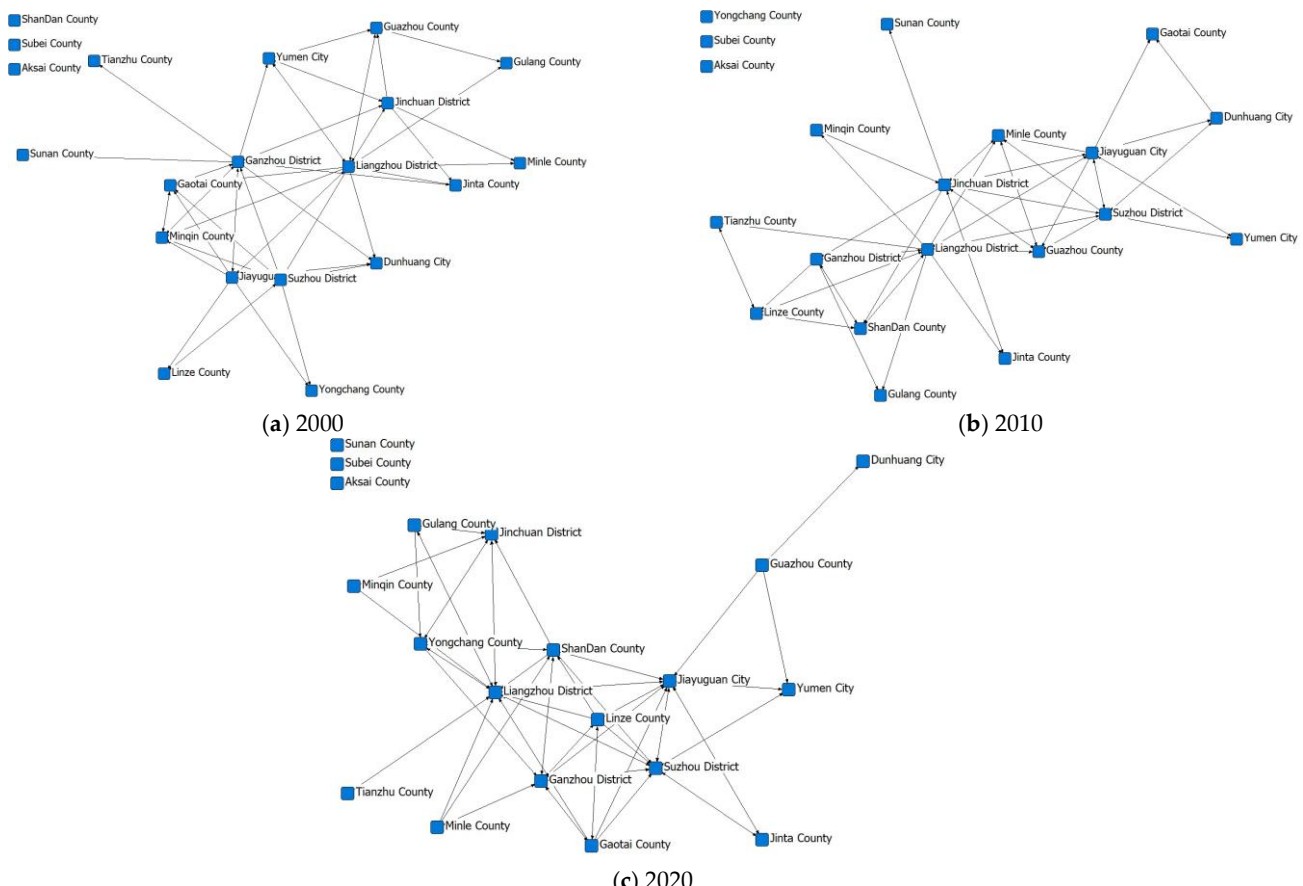

**Figure 6.** The economic spatial network structure of the Hexi Corridor urban agglomeration.

### 3.4.1. Network Density Analysis

The network density reflects the overall development characteristics of the economic connection network [37]. The higher the network density value, the stronger the reasonable allocation of resources and factor flow ability, and the higher the economic efficiency [38].

Table 6 shows that from 2000 to 2020, the network's density doubled, and the regional interactions increased rapidly. From 2000 to 2010, the network density increased from 0.939 to 2.401. Under the background of the western development strategy, Gansu Province swiftly put forward the strategy of "reconstructing Hexi", implemented a comprehensive management policy for the inland river basin, promoted the development of traditional agriculture to modern agriculture, accelerated the development of industrialization, improved the comprehensive strength of cities and towns, and enhanced the economic connection accordingly. In 2020, the network density rose to 7.326. Under the implementation of the Belt and Road Initiative and the National New Urbanization Plan (2014–2020), modern transportation and logistics channels, modern energy channels, and modern information channels were built successively, and the network agglomeration capacity continued to rise.

**Table 6.** The density value of the economic connection network of the Hexi Corridor city group.

| Year | Density of Network |
|------|--------------------|
| 2000 | 0.939 |
| 2010 | 2.401 |
| 2020 | 7.326 |

### 3.4.2. Network Centrality Analysis

In this study, we used the Arc GIS inverse distance weight interpolation method to visualize the spatial representation of the degree centrality (DC), betweenness centrality (BC), and closeness centrality (CC) in 2000, 2010, and 2020 (Figure 7). As we can see, the DC of each city has no significant change from 2000 to 2020. Suzhou District–Jiayuguan District, Ganzhou District, and Liangzhou District are always in the core position. The spatial distribution trend of the economic core area basically coincides with the main traffic route of the Lanzhou–Xinjiang Railway (high-speed railway) and Lianhuo Expressway, showing a bead-like combination shape. The closeness centralization gradually increases from west to east. The southwest section of the Hexi Corridor is located in a high-altitude area with a small population and a multi-ethnic community. The landforms are mainly those of the Gobi Desert, and the transportation infrastructure is weak, so the CC is low. The economic development mainly depends on the Dunhuang–Guazhou group, with a high degree of urbanization to drive it. The betweenness centralization shows an unbalanced trend, and the polarization characteristics are prominent. Suzhou District, Jiayuguan District, and Liangzhou District serve as the center of communication among various regions. At the same time, Ganzhou District, Linze, Gaotai, Yumen, and other counties and districts act as virtual nodes in the channel connection and do not play a prominent role in media and bonds. The above results show that although the density of the economic network is significantly increased, the spatial network structure is relatively loose, and most cities are still in a weak connectivity state.

We conducted a further comparison of the out-degree and in-degree centrality (Figure 8). As seen from Figure 6, the in-degree centrality is greater than the out-degree centrality in the five central urban areas. The out-degree centrality is greater than the in-degree centrality in counties and districts close to the central urban area, such as Linze, Gaotai, Yongchang, and Guazhou. This indicates that the economic resource elements of the urban agglomeration flow from 15 counties and cities to the central urban area, and the agglomeration effect of the economic network is greater than the diffusion effect. Shandan, Linze, Gaotai, Minle, and Tianzhu Counties had higher out-closeness centrality than in-closeness centrality, indicating that the absorption channels of these counties, when

accepting diffusion, were not as smooth as the channels of external diffusion. Although they had good conditions for external communication, they lacked the ability of resource aggregation. The comprehensive strength of counties and districts needed to be improved, leading to the weak role of media.

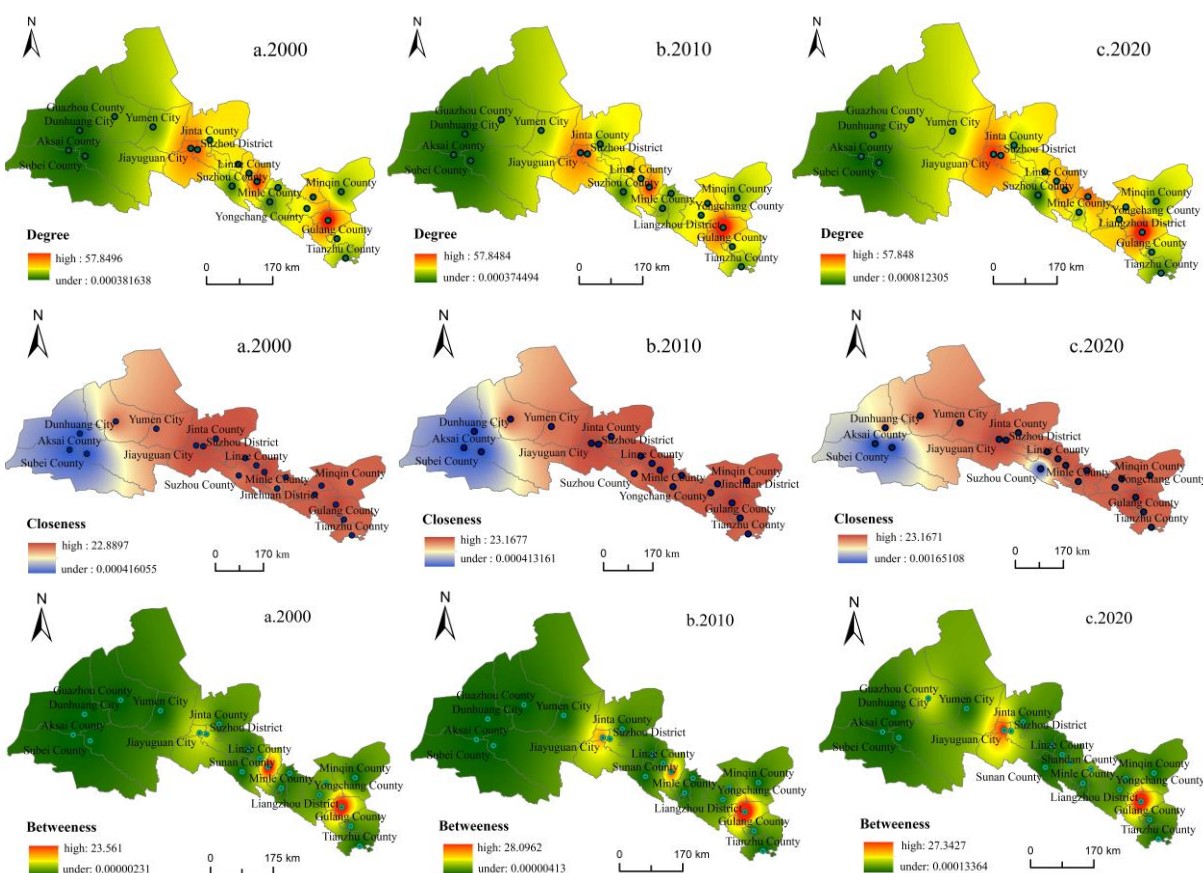

**Figure 7.** Spatial distribution of centrality of the Hexi Corridor in 2000, 2010, and 2020.

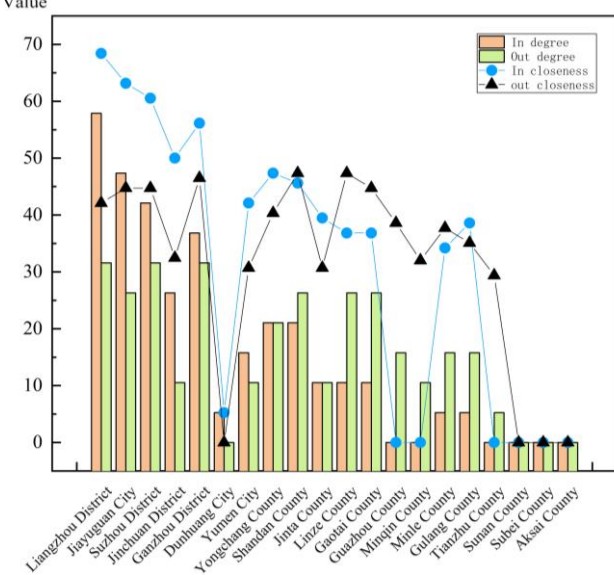

**Figure 8.** Out- and in-degree of DC and CC of the urban agglomeration in 2020.

### 3.4.3. Cohesive Subgroup Analysis

The development of urban agglomerations has roughly experienced four stages, namely resource concentration, driven development, cluster development, and mature development. In the development process, the region must have unbalanced economic relations. Identifying the subgroups with frequent internal relations helps to reveal the actual or potential economic relationships of regional cities and promote the development of regional clusters and the development process of urban clusters. In this paper, we used the convergent correlation (CONCOR) tool of UCINET to cluster the internal structure of the economic network of the urban agglomeration, and we obtained a cohesive subgroup tree diagram (Figure 9).

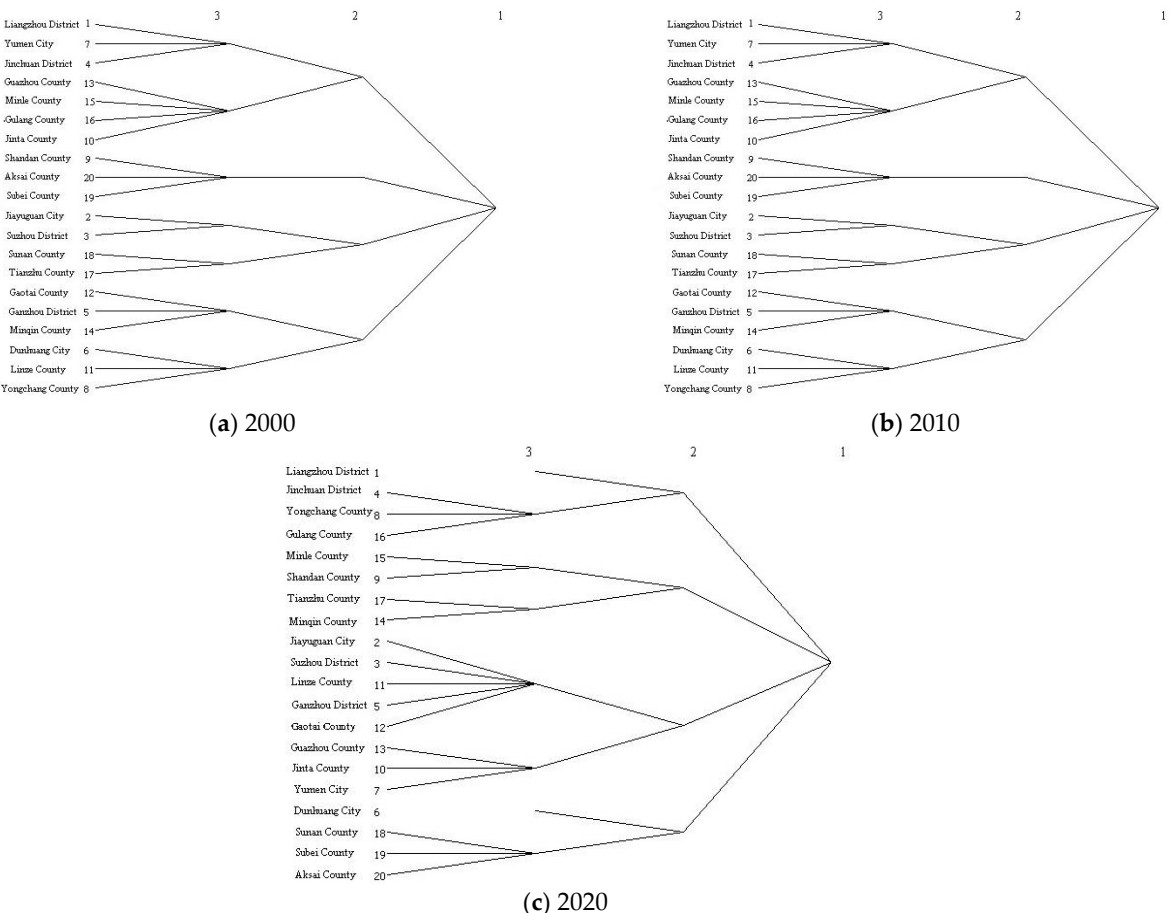

**Figure 9.** Cohesive subgroup.

It can be seen in the tree diagram of the cohesive subgroups in 2000 and 2020 (Figure 9) that the agglomerated subgroups within the Hexi Corridor urban agglomeration are mainly composed of six third-level subgroups and four second-level subgroups, showing a "pyramid"-type distribution. The cohesive subgroups within the Hexi Corridor urban agglomeration in 2000–2020 are mainly composed of seven tertiary clusters and four secondary subgroups, showing a "pyramidal" distribution characteristic. Two subgroups, Jiuquan–Jiayuguan–Zhangye and Jinchang–Wuwei, are more closely structured. The Zhangye and Jiujia regions are divided into the same subgroup due to the high number of connections, which, to a certain extent, indicates the direction of the future economic connection trend of the Zhangye cluster. Integrated urban development plans based on geographical proximity and the breaking down of administrative barriers are gradually becoming an endogenous driving force for changes in the intensity and structure of urban linkages within urban agglomerations.

## 4. Discussion

The uniqueness of the geographical location of the Hexi Corridor and the urban attributes of oasis towns in the arid area determine the oasis agglomeration of five cities in the Hexi Corridor along three inland river basins, forming a relatively independent geographical unit [39]. The fragmentation of the spatial distance and the similar resource advantages within the cluster result in the limited economic interconnection among the clusters of urban agglomerations, the single industrial structure, the primary product level, and the short industrial chain, which make it challenging to form systematic and clustered development. At the same time, the industrial system, dominated by the petrochemical industry, nonferrous metals, and equipment manufacturing, and historical reasons, such as resource monopoly under the existing ownership structure, have caused the structural inertia of resource-based state-owned enterprises such as the Jiugang and Jinchuan Groups. The competitiveness of the characteristic processing enterprises and clusters of agricultural products in Zhangye, Wuwei, and Jiuquan is not strong, and the economic development shows an apparent dual structure [4]. In addition, under the guidance of the western development strategy, the strategy of "Recreating Hexi" in Gansu Province has promoted the accelerated development of modern agriculture and industrialization in the Hexi Corridor, and initially laid the foundation for high-quality development. During the 12th Five-Year Plan period, the "Jiuquan–Jiayuguan Integrated Development Plan" and "Jinchang–Wuwei Integrated Development Plan" effectively promoted the cross-administrative infrastructure construction, the linkage of ecological environment protection, the optimization of the urban development spatial layout, and the promotion of the integration strategy and policy, making full use of the advantages of the urban agglomeration. During the 13th Five-Year Plan period, relying on the "Belt and Road" Initiative, the National Plan for New Urbanization (2014–2020), highspeed railways, expressways, and aviation and other modern transportation and logistics channels, all cities and towns gradually moved towards the strategic frontier of the country's westward opening. The quality and efficiency of the industry have been improved, the tertiary industry is booming, and the economic development is transforming from scale and speed to quality and efficiency. The network of economic links is becoming increasingly complex [11]. However, the overall level of connection between counties in the urban agglomeration is still low, and the economic network structure is mainly driven by the central city of each group and its neighboring counties, which accords with the urban spatial layout characteristics of "large dispersion and small agglomeration" in the Hexi Corridor. At this time, it is particularly important to establish denser transportation lines. According to the "14th Five-Year Plan" Development Plan of the Hexi Corridor Economic Belt, we should build a comprehensive transportation network of "one axis through and multi-dimensional sudden advance", seize the opening opportunities brought by the new round of the western development strategy and the international trade channel opened by the China–Europe Railway Express, and actively communicate with other countries for development. We should also actively participate in the construction of demonstration projects of strategic connectivity in Central Asia, West Asia, and South Asia, jointly hold relevant investment and trade fairs and international exhibitions, strengthen strategic cooperation with mainstream media, establish a new image of open and cooperative development of the Hexi Corridor, and inject new vitality into the development of the urban agglomeration under the current development orientation of tighter resource constraints and ecological goals. Moreover, the leapfrog development of urban agglomerations in underdeveloped areas should be realized [40]. Limited to data acquisition and research methods, this paper focuses on the analysis of the internal economic relation of urban agglomerations, and fails to fully reflect the structure of the economic relations network. The future economic relations with the external space are an important aspect that needs to be further studied. Combined with the research results of this paper, the territorial spatial planning concept of "water source protection in the south, sand control in the north, oasis construction in the middle, balanced and dislocated development of river basin" in the Hexi Corridor, and the functional positioning of the

strategic corridor, the economic cooperative development model focusing on economic agglomeration, regional connectivity, and policy cooperative efficiency [19] will help the banded urban agglomeration of the Hexi Corridor to overcome the geographical and spatial restrictions. We should make full use of the advantages of local resources, form an industrial gradient layout of dislocation and complementarity, realize the integrated development of the industrial chain of new nonferrous metal materials, new chemical materials, new energy, and resource utilization, and transform into an economy based on quality and efficiency. Therefore, according to the "14th Five-Year Plan" Economic Belt Development Plan of the Hexi Corridor, this paper puts forward the economic coordinated and integrated development mode of "one axis, four zones, multiple points, and multiple channels" from the three levels of point, axis, and surface to improve the overall competitiveness of the urban agglomeration (Figure 10).

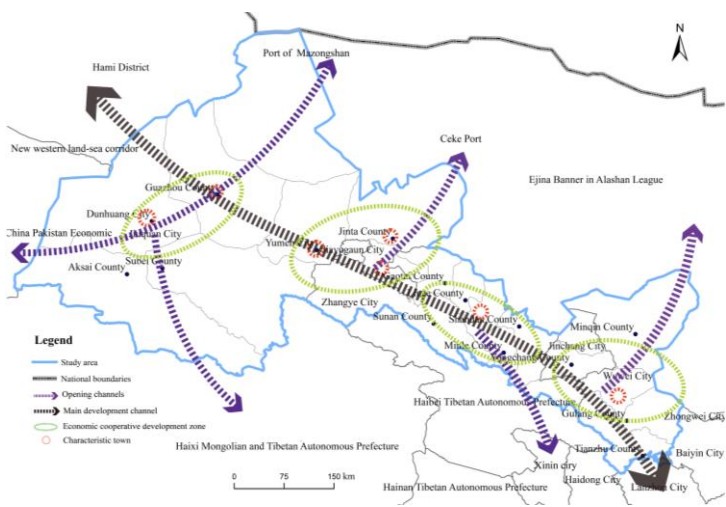

**Figure 10.** Spatial collaborative and integrated development model of the Hexi Corridor urban agglomeration.

Area development model construction. According to the results of the network centrality, the city cluster forms four core economic circles, namely Jiujia, Dungua, Zhangye, and Jinwu. Considering the bearing capacity of resources and the environment and integrating valuable industrial resources, four economic synergistic development zones can be built in the future by developing Jinchang–Wuweis' new nonferrous metal materials and ecological culture, Ganzhou–Linze Gaotai–Minle's emerging industries and ecological culture, Dunhuang–Guazhou's historical and cultural tourism and ecological economy, and Jiuquan–Jiayuguan's new energy and historical culture to promote regionally differentiated and synergistic development.

Axis development model construction. The Hexi Corridor should be considered through the city group of the Lanzhou–Xinjiang Railway, Lian–Huo Expressway, 312 National Road, and other transportation corridors to enhance the development of the central axis, give full play to the advantages of Jiujia and the Wuwei Dry Port, and expand the north–south multi-way open channel to enhance the level of interconnection of the city group [41]. At the same time, to the south and east, the new land and sea trade channel should be expanded to the west, a new Asia–Europe Continental Bridge channel should be built to the south, the China–Pakistan Economic Corridor should be connected [42] to the north, and Jiuquan should be linked to the China–Mongolia ports (Mazongshan Port and Ceke Port), connecting to important node cities and resource concentration areas inside and outside the region.

Multi-point coordinated characteristic development mode. The majority of small and medium-sized towns in the Hexi Corridor urban agglomeration should be included in point development to enhance the characteristic visibility and influence of small towns, which is conducive to strengthening the node-supporting role of cities and towns and

integrating them into the urban economic circle. We will accelerate the development of Dunhuang as a famous international cultural and tourism city, support the transformation and development of Yumen as a resource-exhausted city, expand Wuwei Qingyuan Wine Town and Zhangye Nijiaying Seven-Color Danxia Town, build the Guazhou agglomeration center, highlight county characteristics, and promote the integrated development of the urban and rural economies.

## 5. Conclusions

This paper analyzes the industrial functional connection of the Hexi Corridor urban agglomeration through the urban flow intensity and industrial functional complementarity index, calculates the economic connection intensity based on the revised gravity model, and uses social network analysis to explore the evolution of the economic network structure, drawing the following conclusions.

(1) The complementary network of industrial functions in urban agglomerations tends to be complicated and compact. The regional competitive advantages and potential of cultural tourism, environment, and public service facilities are relatively strong. Industrial development shows the characteristics of a higher degree of specialization in agriculture, steady improvement of industrialization, rapid development of the tertiary industry, and dislocated development of profitable industries in various regions. (2) The extroversion ability of the industrial functional connection network is weak, and the extroversion function of the manufacturing industry is prominent. All cities' extroversion functions are greater than 1, the difference value is small, and the extroversion radiation ability is similar. (3) The primary industry's industrial core backbone connection axis is mainly based on the closed triangular connection between Jiuquan, Zhangye, and Jinchang. The second industry is the Jiujia–Jinwu dual-core linkage, with Jiujia as the lead and Jiuzhang and Jinwu as the backbone. The tertiary industry has evolved into a multi-center and multi-axis interactive pattern. (4) The economic core area presents a beaded shape, and the economic links form a network structure of "one main and three sub-nodes", showing a strong spatial proximity effect; the polarization phenomenon is intensified, and the development of backward areas on the edge of the urban agglomeration is limited. (5) The structure of Jiuquan–Jiayuguan–Zhangye and Jinchang–Wuwei subgroups is close. Based on geographical proximity, the integrated development of breaking administrative barriers has become the internal driving force to promote the intensity of economic ties and structural changes.

**Author Contributions:** Conceptualization, X.X. and P.S.; methodology, X.X.; software, X.X.; formal analysis, X.X.; writing—original draft preparation, X.X.; writing—review and editing, P.S. and X.X.; visualization, X.X.; funding acquisition, P.S. All authors have read and agreed to the published version of the manuscript.

**Funding:** This research was funded by the National Natural Science Foundation of China, No. 41771130.

**Data Availability Statement:** The socio-economic data are derived from the Statistical Yearbook of the Cities of Hexi Corridor Urban Agglomeration in 2001, 2011, and 2021; the Gansu Development Yearbook; the China County Statistical Yearbook; the China City Statistical Yearbook, and the CSMAR Economic Geographic Database. The freight volumes of roads and railways come from the annual statistical reports of each city's national economic and social development. The railway time cost is derived from the China railway customer service center website (https://kyfw.12306.cn/otn/left-Ticket/init, accessed on 27 March 2022). Highway time selection of highway costs and the shortest time distance are derived from the official website of Baidu maps (http://map.baidu.com/, accessed on 27 March 2022). The national map is based on the standard map No. GS(2019)1824, approved by the Ministry of Natural Resources of China. The base map is not modified. Some or all data, models, or codes that support the findings of this study are available from the corresponding author upon reasonable request.

**Acknowledgments:** We gratefully acknowledge the support of the National Natural Science Foundation of China, No. 41771130.

**Conflicts of Interest:** The authors declare no conflict of interest.

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
