# Peer review of "Dynamic Evolution and Collaborative Development Model of Urban Agglomeration in Hexi Corridor from the Perspective of Economic Flow"

_land, doi:10.3390/land12020274_

Round 1

Reviewer 1 Report

The manuscript addresses a crucial issue regarding regional development in terms of economic flow. The work is very interesting.

In the introduction, the authors could have referred to the review of the literature to a greater extent, and in particular, the economic factors shaping the strength of regional ties. Why Discussion is after the Conclusion? The separation of guidance for shareholders is also missing. The chapters were numbered incorrectly.

Author Response

Point 1: In the introduction, the authors could have referred to the review of the literature to a greater extent, and in particular, the economic factors shaping the strength of regional ties.

Response 1: Please provide your response for Point 1. (in red)

Dear Reviewers,  thank you very much for your question regarding the appearance of the literature review in the introduction. After analyzing the issues you raised, I have re-collected and re-organized the research content of the relevant literature to add the methods and research content of economic linkage research in different periods at home and abroad in a chronological structure, in order to be able to describe the current progress of economic linkage research more accurately and well. Please review it again, and I look forward to your reply.

Point 2: Why Discussion is after the Conclusion? The separation of guidance for shareholders is also missing.

Response 2: Please provide your response for Point 2. (in red)

Dear Reviewer, Thank you very much for your questions, which have helped me to further understand the structure of the paper. I have set the discussion section before the conclusion. I would appreciate your approval and thank you again for your review comments.

Point 3: The chapters were numbered incorrectly.

Response 3: Please provide your response for Point 3. (in red)

Dear reviewers, Thank you very much for your questions. I have corrected the incorrect chapter numbering and have ensured the correct chapter numbering for your approval.

Reviewer 2 Report

It is quite an interesting topic about  the industrial function connection path and economic network characteristics of Hexi Corridor through the urban flow model. Here are some suggestions for the manuscript improvement.

First, it would be better to present the research significance of the study at the end of the abstract.

Then, in the introduction part the literature review is inadquate since the international studies were not included.

In addition,  discussion part should follow the results (part3).  It is better to to discuss" Coordinated Economic Development Mode in Hexi Corridor " and its applicability and generalization  in the discussion part.  

Author Response

Point 1:  it would be better to present the research significance of the study at the end of the abstract.

Response 1: Please provide your response for Point 1. (in red)

Dear Reviewers, thank you very much for your advice on abstract writing. According to your suggestion, I have added the research significance of this paper to the abstract. The modified content is as follows:

 It is of great significance to strengthen the urban connection in Hexi Corridor and give full play to the overall competitive advantage. 

Point 2: in the introduction part the literature review is inadquate since the international studies were not included.

Response 2: Please provide your response for Point 2. (in red)

Dear Reviewer, thank you very much for your advice on the introduction. After fully thinking about the questions you raised, I sort out the research content of economic connection at home and abroad, and summarize the research content and research methods of economic connection in chronological order, and supplement the research content of other international scholars to enrich the introduction. The introduction has now been rewritten.

Point 3:  discussion part should follow the results (part3).  It is better to to discuss" Coordinated Economic Development Mode in Hexi Corridor " and its applicability and generalization  in the discussion part.  

Response 3: Please provide your response for Point 3. (in red)

Dear reviewers, Thank you very much for your suggestions on the structure of this paper. At present, I have put "Hexi Corridor economic integration and collaborative development model" into the discussion content to make the structure of this paper clearer.

Reviewer 3 Report

The topic of the paper is relevant and very interesting. Authors used a wide range of related literature sources and cited them correctly. Most part of them are from the last few years. This paper analyses the industrial functional connection of the Hexi Corridor urban agglomeration through the urban flow intensity and industrial function complementarity index, calculates the economic connection intensity. It based on a wide range of and strong methodologies such as the revised gravity model and uses social network analysis to explore the evolution of the economic network structure, based on the time series of 2000, 2010, 2020. These methods supported authors to reach valuable results which were presented in their article. I appreciate them.

I have only one technical suggestion which would be necessary to follow. In the paper I could find two chapters which were signed by number 3.

”3. Methods”, below it the next subchapters number is “2.1 Urban Flow Intensity Model”.

“3. Results”

Author Response

Point 1: The chapter number of the paper is wrong.

Response 1: Please provide your response for Point 1. (in red)

Dear Reviewers, Thank you very much for your suggestion on the article content numbering. After modification, I have numbered the article subheadings in order to ensure that there will be no such structural numbering problems.

Reviewer 4 Report

Interesting and advanced research methods were used in the research described in the article. It also seems that these methods have been applied correctly and the data looks reliable. Unfortunately, the research gap was not indicated, nor was it presented to what extent this gap was filled with new knowledge obtained as a result of research. The purpose of the research has not been specified anywhere, nor has it been explained why and for whom this purpose is important. In other words, it is not known to whom and for what this research may be useful. Therefore, it is very difficult to assess their value. Perhaps these are valuable studies that bring completely new, groundbreaking knowledge to science. Perhaps this knowledge will help to solve some important practical problems in China or in other countries. Unfortunately, the article in its current form does not give you the opportunity to evaluate it.

Author Response

Point 1: the research gap was not indicated, nor was it presented to what extent this gap was filled with new knowledge obtained as a result of research. The purpose of the research has not been specified anywhere, nor has it been explained why and for whom this purpose is important. In other words, it is not known to whom and for what this research may be useful.

Response 1: 

Dear reviewers, thank you very much for your questions on this article. For your question, I would like to explain as follows: As an important fulcrum of the smooth revival of the Silk Road Economic Belt, the unique geographical spatial oasis distribution of Hexi Corridor makes it form a relatively independent cluster economic spatial pattern dominated by the central towns of the oasis. For a long time, the competitiveness of industrial clusters is weak, the degree of internal and external openness is not high, and the integration and coordination development is slow. In order to find out the industrial connection path, urban function connection and county-district economic connection of urban clusters, this paper studies from the perspective of economic flow, combines the intensity of urban flow, the modified gravity model and the analysis method of industrial advantage flow, and makes up for the deficiency of single county-level scale or city-level scale. At the same time, the previous studies focused on the intensity of economic ties and spatial pattern to describe the coordination of urban industrial functions in a vague way. Finally, combined with the functional positioning of the Hexi Corridor, this paper puts forward the coordinated and integrated development model of economic space of urban interconnection, interaction and complementation in the region. It plays an important role in understanding the status quo of regional economic ties, improving the efficiency of resource allocation within urban clusters, enhancing the internal economic ties of underdeveloped urban clusters along the Belt and Road, and promoting the high-quality development of regional integration. After analyzing the questions you raised, I further sorted out the content and significance of the research in the introduction, in order to be able to explain the purpose, significance and innovation of the research. The modified content is as follows. Please review it again and look forward to your reply.

To sum up, this paper chooses the Hexi Corridor urban agglomeration as the research area, which is at the forefront of the opening-up of "The Belt and Road" and located in an economically underdeveloped region with an extremely important geographical location. From the perspective of economic flow, combined with the intensity of urban flow and the revised gravity model, the specific industrial connection path at the municipal level and the economic spatial connection network between counties and cities within the urban agglomeration are clarified, which makes up for the deficiencies of the research scale of a single county or urban area and the specific industrial connection path. Finally, combined with the functional positioning of the Hexi Corridor, this paper proposes a coordinated and integrated development mode of economic space featuring interconnection, interaction, and complementarity among cities within the region, so as to provide a scientific reference for improving the efficiency of resource allocation in urban agglomerations, enhacing the internal economic connectivity of underdeveloped urban agglomerations along the Belt and Road, and promoting the high-quality development of regional integration.

Point 2: Extensive editing of English language and style required

Response 2: 

Dear reviewer, I have edited the full text in English. Please review it again, and I look forward to your reply.

Point 3: The research design needs to be improved

Response 3: 

Dear reviewer, I have combined the research area overview and technology roadmap in Section 2 with the research method in Section 3 and revised the structure of the article. Please review it again.

That's all I've revised. Thank you again for your advice and looking forward to your reply.

Round 2

Reviewer 1 Report

I accept in present form.

Author Response

Thank you for your review

Reviewer 2 Report

The revised version is quite good.

Author Response

Thank you for your review

Reviewer 4 Report

Corrected version of the article meets the requirements